# Optimal Decoding Order and Power Allocation for Sum Throughput Maximization in Downlink NOMA Systems

**DOI:** 10.3390/e26050421

**Published:** 2024-05-15

**Authors:** Zhuo Han, Wanming Hao, Zhiqing Tang, Shouyi Yang

**Affiliations:** 1School of Electrical and Information Engineering, Zhengzhou University, Zhengzhou 450001, China; sincerezhuohan@sina.com (Z.H.); iesyyang@zzu.edu.cn (S.Y.); 2School of Cyber Science and Engineering, Zhengzhou University, Zhengzhou 450001, China; iezqtang@zzu.edu.cn

**Keywords:** NOMA, decoding order, power allocation, outage probability, sum throughput

## Abstract

In this paper, we consider a downlink non-orthogonal multiple access (NOMA) system over Nakagami-*m* channels. The single-antenna base station serves two single-antenna NOMA users based on statistical channel state information (CSI). We derive the closed-form expression of the exact outage probability under a given decoding order, and we also deduce the asymptotic outage probability and diversity order in a high-SNR regime. Then, we analyze all the possible power allocation ranges and theoretically prove the optimal power allocation range under the corresponding decoding order. The demarcation points of the optimal power allocation ranges are affected by target data rates and total power, without an effect from the CSI. In particular, the values of the demarcation points are proportional to the total power. Furthermore, we formulate a joint decoding order and power allocation optimization problem to maximize the sum throughput, which is solved by efficiently searching in our obtained optimal power allocation ranges. Finally, Monte Carlo simulations are conducted to confirm the accuracy of our derived exact outage probability. Numerical results show the accuracy of our deduced demarcation points of the optimal power allocation ranges. And the optimal decoding order is not constant at different total transmit power levels.

## 1. Introduction

Non-orthogonal multiple access (NOMA) is a potential candidate for a future multiple access technique. It was proposed by NTT DoCoMo in 2013, aiming at addressing the massive connectivity issue in wireless communication systems. NOMA has a good capacity in improving spectrum efficiency through sharing the same time/frequency/code resources, so it offers a better capacity performance compared with that of orthogonal multiple access (OMA). In power-domain NOMA, desired signals for different users with different power levels are combined using superposition coding (SC) at the transmitter, and each user decodes its own desired signal through successive interference cancellation (SIC). We refer to power-domain NOMA as NOMA in the rest of the paper.

### 1.1. Related Works

The performance of NOMA depends on the power allocation under the SC technique and the decoding order using the SIC technique. In earlier studies on NOMA systems, the SIC decoding order was prefixed using the ascending order of the users’ channel gain [1], i.e., the user with the worse channel gain is allocated more power. The drawback of the prefixed SIC decoding order strategy is that it may bring performance degradation under fading channels, because the decoding order cannot vary with the variation in the instantaneous channel state information (CSI). To address this problem, some researchers have proposed a dynamic SIC decoding order strategies based on instantaneous CSI [2,3], instantaneous received power [4], and a ratio of the instantaneous received power to the target data rate [5]. The above-mentioned dynamic strategies can provide sensible decoding orders for fading channel scenarios and avoid exhaustive searching in all the possible decoding orders. However, they do not ensure the provision of the optimal decoding order to make full use of NOMA.

To find the optimal decoding order, some researchers have made efforts in theoretical analyses, given specific scenarios or conditions. For example, ref. [6] proved that the optimal decoding order for a sum rate maximization problem subjected to minimum user rate requirements under perfect CSI is determined using the ascending order of the channel gain. In [7], the optimal decoding order for the sum rate maximization problem under perfect CSI in a NOMA-CoMP system was proved to depend on the differentials of the channel gains. The authors of [8] showed that the optimal decoding order follows the descending order of the channel gain in the joint decoding order and computation resource allocation optimization problem under perfect CSI when the size of task data for two devices are equal. In [9], the authors proposed a necessary condition for a SIC decoding order to be optimal. In [10], it was demonstrated that decoding the far NOMA user first at the BS provides the best performance in two-user uplink (UL) cooperative non-orthogonal multiple access (C-NOMA) cellular networks. Ref. [11] derived a closed-form optimal decoding order for a total transmit power minimization problem subjected to outage constraints under statistical CSI. Ref. [12] derived the optimal decoding order with regard to the max–min fairness problem for a downlink NOMA system under statistical CSI. In studies based on theoretical analysis, refs. [6,7,8,9,10] assumed that perfect CSI is known to the transmitter, while refs. [11,12] considered the statistical CSI. For a majority of optimization problems involving an optimal decoding order, it is usually infeasible to find a closed-form solution through theoretical analysis. And experimental searching is used to find an optimal or suboptimal solution, such as in [13,14,15].

Similar to the optimal decoding order, the finding of optimal power allocation can also be solved through theoretical analysis [16,17,18,19] or experimental searching [20,21]. In [16], two closed-form optimal power allocation solutions were derived based on the Karush– Kuhn–Tucker (KKT) conditions by fixing the strong user’s capacity and the weak user’s capacity, respectively. In [17], the optimal power allocation was derived using KKT conditions because the energy-efficient (EE) maximization problem can be transformed into a convex optimization problem. Ref. [18] obtained the closed-form optimal power allocation for two paired users in an achievable sum secrecy rate maximization problem. Ref. [22] derived the optimal solutions of power allocation depending on which user is the D2D transmitter in the sum rate maximization problem subjected to minimum rate constraints. The authors of [19] derived the closed-form expression of power allocation based on KKT conditions to maximize the total throughput under the constraints of the data rate, sum transmit power, and atomicity of devices’ tasks. It can be seen that a closed-form solution of optimal power allocation in certain optimization problems can only be obtained under special conditions, such as perfect CSI. Ref. [20] introduced a block-successive upper-bound minimization algorithm to find the globally optimal power allocation. Ref. [21] used deep reinforcement learning to solve the power allocation in a throughput maximization problem. By the way, some studies have used prefixed power allocation for its advantages in applications of the internet of things without full CSI [23].

### 1.2. Motivation and Contributions

Optimal solutions about decoding order and power allocation usually do not exist in non-convex optimization problems, or they cannot be found using current theoretical analysis methods. With such situations, experimental searching plays a part. A variety of searching algorithms have been designed to find the optimal decoding order and power allocation. However, these searching algorithms mainly conduct searching in the whole domain of the defined problem. When the number of users increases, the possible decoding orders and power allocation ranges also increase rapidly. The problems of algorithm inefficiency resulting from searching in the whole definition domain become more and more obvious. If the well-designed searching algorithms can search in a reduced power allocation range, then they will become more efficient in solving the joint decoding order and power allocation optimization problem by avoiding searching in useless ranges. To the best of our knowledge, the attainment of an optimal power allocation range through in-depth theoretical analysis to reduce the searching space in NOMA systems considering statistical CSI is still an open problem. This was the main motivation for us to carry out this study. We aimed to provided a reduced power allocation range to improve the searching efficiency of the well-designed searching algorithms, including machine learning algorithms, by avoiding searching in useless ranges. In this study, we considered a downlink NOMA system over Nakagami-*m* fading channels, where each node (one BS and multiple NOMA users) is equipped with a single antenna. We assumed that only statistical CSI is available to the BS. It is quite costly for the BS to achieve perfect CSI in practice, because wireless channels may vary randomly due to terminal mobility, weather variation, stochastic noise, and so on. So, the investigation on NOMA systems based on statistical CSI is of great necessity. A joint optimal decoding order and power allocation problem were formulated to maximize the sum throughput. The main contributions of this study are as follows:We derived a closed-form expression of the exact outage probability and corresponding asymptotic outage probability under given decoding orders. We also deduced the diversity order for insight of the characteristic of the outage probability in a high-signal-to-noise-ratio (high-SNR) regime.We analyzed all the possible power allocation ranges under each decoding order in detail, and we determined the optimal power allocation ranges to reduce the searching space. This is the main novelty of our work. The theoretical analysis results indicate that the demarcation points of the optimal power allocation ranges are affected by the target data rates and total power, without an effect from the CSI. In particular, the values of the demarcation points are proportional to the total power.We formulated a joint decoding order and power allocation optimization problem to maximize the sum throughput, which is solved by efficiently searching in the obtained optimal power allocation ranges.We conducted Monte Carlo simulations to confirm the accuracy of our derived exact outage probability. The numerical results show the accuracy of our deduced demarcation points of the optimal power allocation ranges. And we found that the optimal decoding order is not constant at different total transmit power levels.

The remainder of this paper is organized as follows. Section 2 describes the system model. Section 3 presents the outage probability analysis. Section 4 displays the power allocation analysis. Section 5 shows the formulation of the optimization problem. Section 6 shows the simulation results. Section 7 shows the discussion. Finally, Section 8 concludes this paper.

## 2. System Model

We consider a single-cell downlink NOMA system with a base station (BS) and *K* users, where each node is equipped with a single antenna. We assumed that the channel between the BS and each user experiences quasi-static Nakagami-*m* block fading. A Nakagami-*m* channel is a universal fading channel model, where parameter *m* defines the fading shape, and Ω is the average channel gain. In particular, Rayleigh channel is a special case of a Nakagami-m channel with m=1. The CSI remains constant during one block but changes independently and randomly from one block to the next block. The BS only knows the long-term statistical CSI associated with each user, because it is costly to achieve perfect CSI via continual channel feedback. It is worth noting that the BS only conducts one-time power allocation over all the blocks, i.e., the power allocation factors remain unchanged as long as the statistical CSI of two users remains unchanged.

The signal received by user $U_{k}$ can be expressed as
(1)$$\begin{array}{c} y_{k}=h_{k}\left(\sum_{k=1}^{K}\sqrt{a_{k}P}\;x_{k}\right)+n_{k}, \end{array}$$
where k=1,2,…,K, and $h_{k}∼\mathrm{Nak}\left(m_{k},\Omega_{k}\right)$ denotes the channel coefficient of user $U_{k}$. $x_{k}$ denotes the desired signal for user $U_{k}$, where $E[x_{k}]=0$, $E[x_{k}^{2}]=1$. $a_{k}$ is the power allocation factor of user $U_{k}$, and $\sum_{k=1}^{K}a_{k}=1$. *P* is the total transmission power for NOMA users. $n_{k}$ is the additive white Gaussian noise (AWGN) with a zero mean and variance $\sigma_{k}^{2}$. The cumulative distribution function (CDF) of channel gain $g_{k}=\left|h_{k}^{2}\right|$ can be expressed as $F\left(g_{k}\right)=\gamma \left(m_{k},\frac{m_{k}}{\Omega_{k}}g_{k}\right)/\Gamma \left(m_{k}\right)$. Γ(m) is the gamma function, which can be calculated by gamma(m) in Matlab R2014a and more recent versions. $\gamma \left(m,x\right)=\int_{0}^{x}t^{m-1}e^{-t}dt$ denotes the lower incomplete gamma function, which can be calculated by gammainc(x,m)∗gamma(m) in Matlab R2014a and more recent versions.

Let π=(π(1),π(2),…,π(K)) denote a possible decoding order of the desired signals, where π(j)=k means that signal $x_{k}$ is the *j*-th one to be decoded, j=1,2,…,K. According to the principle of the NOMA scheme, user $U_{\pi (i))}$ is able to subtract signal $x_{\pi (l))}$ using the SIC technique, l≤i, while user $U_{\pi (i)}$ cannot subtract signal $x_{\pi (j)}$, i<j, which is treated as interference noise. Let $R_{\pi (l)}^{\pi (i)}$ denote the instantaneous data rate that user $U_{\pi (i)}$ can determine about the desired signal $x_{\pi (l)}$ when using the SIC technique, where l≤i. Then, we have [1] ([Equation (Equation 3)])
(2)$$\begin{array}{c} R_{\pi (l)}^{\pi (i)}=\log_{2}\left(1+\frac{g_{\pi (i)}a_{\pi (l)}P}{g_{\pi (i)}\sum_{j=l+1}^{K}a_{\pi (j)}P+\sigma_{\pi (i)}^{2}}\right). \end{array}$$

## 3. Outage Probability Analysis

In NOMA systems, a desired signal can be successfully decoded only when *the instantaneous data rate is no less than its target data rate* [1]. The outage probability reveals the probability that a user-desired signal cannot be successfully decoded at its own receiver.

### 3.1. Exact Outage Probability

Let $R_{\pi (i)}^{F}$ (in bits/s/Hz) denote the fixed target data rate of user $U_{\pi (i)}$, which is a predetermined parameter. We define event $\left\{R_{\pi (l)}^{\pi (i)}\geq R_{\pi (l)}^{F}\right\}$ as user $U_{\pi (i)}$ being able to decode signal $x_{\pi (l)}$, l≤i. The outage probability of user $U_{\pi (i)}$ can be defined as
(3)$$\begin{array}{c} P_{\pi (i)}^{\mathrm{out}}≜1-\mathrm{Pr}\left[\underset{l\leq k}{\cap }\left\{R_{\pi (l)}^{\pi (i)}\geq R_{\pi (l)}^{F}\right\}\right]. \end{array}$$

Let $\xi_{\pi (l)}\;=\;2^{R_{\pi (l)}^{F}}\;-\;1$. Then, $R_{\pi (l)}^{\pi (i)}\geq R_{\pi (l)}^{F}$ can be rewritten as
(4)$$\begin{array}{c} g_{\pi (i)}\geq \left(\xi_{\pi (l)}\sigma_{\pi (i)}^{2}/P\right)/\left(a_{\pi (l)}-\xi_{\pi (l)}\sum_{j=l+1}^{K}a_{\pi (j)}\right) \end{array}$$

**Theorem 1.** 
*Let ${\bar{\gamma }}_{\pi (i)}=P/\sigma_{\pi (i)}^{2}$ denote the SNR. The exact outage probability can be written as*

(5)
$$\begin{array}{c} P_{\pi (i)}^{out}=\gamma \left(m_{\pi (i)},\frac{m_{\pi (i)}\eta_{\pi (i)}}{\Omega_{\pi (i)}{\bar{\gamma }}_{\pi (i)}}\right)/\Gamma \left(m_{\pi (i)}\right), \end{array}$$

*where $\eta_{\pi (i)}=\max\left\{\frac{\xi_{\pi (l)}}{a_{\pi (l)}-\sum_{j=l+1}^{K}\xi_{\pi (l)}a_{\pi (j)}}\right\},\;i=1,2,\ldots ,K,\;\mathrm{and}\;l=1,2,\ldots ,i.$*


**Proof.** $P_{\pi (i)}^{\mathrm{out}}$ can be directly derived with the aid of $F(g_{\pi (i)})$. □

### 3.2. Asymptotic Outage Probability

The asymptotic outage probability can be used to calculate the exact outage probability in the high-SNR regime. The asymptotic outage probability also reveals the diversity order, which is the slope of the asymptotic line of exact outage probability.

**Theorem 2.** 
*We write the exact outage probability in *(Equation 5)* in a particular form as $P_{\pi (i)}^{out}={\left(\Phi_{\pi (i)}{\bar{\gamma }}_{\pi (i)}\right)}^{-\delta_{\pi (k)}}+o\left({\bar{\gamma }}_{\pi (i)}^{-\delta_{\pi (k)}}\right)$. When ${\bar{\gamma }}_{\pi (i)}\to +\infty$, the remainder term $o({\bar{\gamma }}_{\pi (i)}^{-\delta_{\pi (i)}})$ can be omitted. The asymptotic outage probability can be obtained using*

(6)
Pπ(i)out∞≈Φπ(i)γ¯π(i)−δπ(i),

*where $\Phi_{\pi (i)}={\left(\Gamma (m_{\pi (i)}+1)\right)}^{\frac{1}{m_{\pi (i)}}}{\left(\frac{m_{\pi (i)}\eta_{\pi (i)}}{\Omega_{\pi (i)}}\right)}^{-1}$*,* and $\delta_{\pi (i)}=m_{\pi (i)}$ is the diversity order.*


**Proof.** Let $C_{\pi (i)}=\frac{m_{\pi (i)}\eta_{\pi (i)}}{\Omega_{\pi (i)}}$. According to the series representation $\gamma \left(\alpha ,x\right)=\sum_{n=0}^{\infty }\frac{{\left(-1\right)}^{n}x^{\alpha +n}}{n!(\alpha +n)}$, we can rewrite (Equation 5) as $P_{\pi (i)}^{\mathrm{out}}=\left[{\left({C_{\pi (i)}{\bar{\gamma }}_{\pi (i)}}^{-1}\right)}^{m_{\pi (i)}}\sum_{n=0}^{\infty }\frac{{\left(-1\right)}^{n}{\left({C_{\pi (i)}{\bar{\gamma }}_{\pi (i)}}^{-1}\right)}^{m_{\pi (i)}+n}}{n!(m_{\pi (i)}+n)}\right]/\Gamma (m_{\pi (i)})\;=\left[\frac{{\left({C_{\pi (i)}{\bar{\gamma }}_{\pi (i)}}^{-1}\right)}^{m_{\pi (i)}}}{m_{\pi (i)}}+\sum_{n=1}^{\infty }\frac{{\left(-1\right)}^{n}{\left({C_{\pi (i)}{\bar{\gamma }}_{\pi (i)}}^{-1}\right)}^{m_{\pi (i)}+n}}{n!(m_{\pi (i)}+n)}\right]/\Gamma (m_{\pi (i)})\;={\left({C_{\pi (i)}{\bar{\gamma }}_{\pi (i)}}^{-1}\right)}^{m_{\pi (i)}}/\Gamma \left(m_{\pi (i)}+1\right)+o\left({\bar{\gamma }}_{\pi (i)}^{-m_{\pi (i)}}\right)={\left[\Gamma {\left(m_{\pi (i)}+1\right)}^{\frac{1}{m_{\pi (i)}}}{C_{\pi (i)}}^{-1}{\bar{\gamma }}_{\pi (i)}\right]}^{-m_{\pi (i)}}+\;o\left({\bar{\gamma }}_{\pi (i)}^{-m_{\pi (i)}}\right)={\left(\Phi_{\pi (i)}{\bar{\gamma }}_{\pi (i)}\right)}^{-m_{\pi (i)}}+o\left({\bar{\gamma }}_{\pi (i)}^{-m_{\pi (i)}}\right).$ □

### 3.3. Diversity Order

The diversity order is defined as $\delta_{k}≜-\lim_{\bar{\gamma }\to +\infty }\frac{\mathrm{lg}P_{k}^{\mathrm{out}}}{\mathrm{lg}\bar{\gamma }}$. In fact, the diversity order can be indicated by the slope of the asymptotic OP in the high-SNR regime.

**Theorem 3.** 
*The diversity order of a NOMA user can be written as*

(7)
$$\begin{array}{c} \delta_{\pi (i)}=m_{\pi (i)},\;i=1,2,\ldots ,K. \end{array}$$



**Proof.** With the aid of the asymptotic outage probability, we can use L’Hospital’s Rule to obtain the diversity order: $\delta_{\pi (i)}=-\lim_{{\bar{\gamma }}_{\pi (i)}\to +\infty }\frac{\mathrm{lg}P_{\pi (i)}^{\mathrm{out}}}{\mathrm{lg}{\bar{\gamma }}_{\pi (i)}}=-\lim_{{\bar{\gamma }}_{\pi (i)}\to +\infty }\frac{{\mathrm{lg}}^{\infty }P_{\pi (i)}^{\mathrm{out}}}{\mathrm{lg}{\bar{\gamma }}_{\pi (i)}}=-\lim_{{\bar{\gamma }}_{\pi (i)}\to +\infty }\frac{-m_{\pi (i)}\mathrm{lg}{\bar{\gamma }}_{\pi (i)}\;-\;m_{\pi (i)}\mathrm{lg}\Phi_{\pi (i)}}{\mathrm{lg}{\bar{\gamma }}_{\pi (i)}}=m_{\pi (i)}.$ □

## 4. Power Allocation Analysis

Each decoding order has a corresponding power allocation range. The power allocation range originally generated by the decoding order can be further narrowed using our in-depth analysis. The reduced power allocation range is called the optimal power allocation range, which contributes to avoiding invalid searches in useless ranges and reducing the computation time. We take two-user scenario as examples first, and then we expand the analysis to multi-user scenarios.

### 4.1. Two-User NOMA Case

For a two-user scenario, the decoding order set is π={(1,2),(2,1)}, where π=(1,2) and π=(2,1) mean that $U_{1}$ and $U_{2}$ are the first users to be decoded, respectively.

#### 4.1.1. Feasible Power Allocation Range I

We denote the power allocation range I as the feasible power allocation under decoding order π=(1,2). The denominator of $\eta_{\pi (i)}$ in Theorem 1 can be written as $a_{1}-\xi_{1}a_{2}>0,\;a_{2}>0$. Considering $a_{1}+a_{2}=1$, we can obtain the condition for meeting the decoding order π=(1,2) as
(8)$$\begin{array}{c} \left\{a_{1}>\xi_{1}/\left(1+\xi_{1}\right)\right\}\cap \left\{a_{2}<1/\left(1+\xi_{1}\right)\right\} \end{array}$$

With $\eta_{\pi (2)}=\max\left\{\frac{\xi_{1}}{a_{1}-\xi_{1}a_{2}},\;\frac{\xi_{2}}{a_{2}}\right\}$, we divide power allocation range I into two sub-ranges: power allocation range I-A and power allocation range I-B. In power allocation range I-A, $\eta_{\pi (2)}=\frac{\xi_{2}}{a_{2}}$, and the power allocation satisfies the inequalities as follows:(9)$$\begin{array}{c} \left\{\frac{\xi_{1}\left(1+\xi_{2}\right)}{\xi_{1}+\xi_{1}\xi_{2}+\xi_{2}}\leq a_{1}<1\right\}\cap \left\{0<a_{2}\leq \frac{\xi_{2}}{\xi_{1}+\xi_{1}\xi_{2}+\xi_{2}}\right\} \end{array}$$

In power allocation range I-B, $\eta_{\pi (2)}=\frac{\xi_{1}}{a_{1}-\xi_{1}a_{2}}$, and the power allocation satisfies the inequalities as follows:(10)$$\begin{array}{c} \left\{\;\frac{\xi_{1}}{1\;+\;\xi_{1}}<a_{1}<\frac{\xi_{1}\left(1+\xi_{2}\right)}{\xi_{1}\;+\;\xi_{1}\xi_{2}+\;\xi_{2}}\;\right\}\;\cap \;\left\{\;\frac{\xi_{2}}{\xi_{1}\;+\;\xi_{1}\xi_{2}\;+\;\xi_{2}}<a_{2}<\frac{1}{1\;+\;\xi_{1}}\;\right\} \end{array}$$

#### 4.1.2. Feasible Power Allocation Range II

We denote power allocation range II as the feasible power allocation under decoding order π=(2,1). The denominator of $\eta_{\pi (i)}$ in Theorem 1 can be written as $a_{2}-\xi_{2}a_{1}>0$, $a_{1}>0$. Considering $a_{1}+a_{2}=1$, we can obtain the condition meeting decoding order π=(2,1) as follows:(11)$$\begin{array}{c} \left\{a_{2}>\xi_{2}/\left(1+\xi_{2}\right)\right\}\cap \left\{a_{1}<1/\left(1+\xi_{2}\right)\right\} \end{array}$$

With $\eta_{\pi (1)}=\max\left\{\frac{\xi_{2}}{a_{2}-\xi_{2}a_{1}},\;\frac{\xi_{1}}{a_{1}}\right\}$, we divide power allocation range II into two sub-ranges: power allocation range II-A and power allocation range II-B. In power allocation range II-A, $\eta_{\pi (1)}=\frac{\xi_{1}}{a_{1}}$, and the power allocation satisfies the inequalities as follows:(12)$$\begin{array}{c} \left\{\frac{\xi_{2}\left(1+\xi_{1}\right)}{\xi_{1}+\xi_{1}\xi_{2}+\xi_{2}}\leq a_{2}<1\right\}\;\cap \;\left\{0<a_{1}\leq \frac{\xi_{1}}{\xi_{1}+\xi_{1}\xi_{2}+\xi_{2}}\right\} \end{array}$$
In power allocation range II-B, $\eta_{\pi (1)}=\frac{\xi_{2}}{a_{2}-\xi_{2}a_{1}}$, the power allocation satisfies the inequalities as follows:(13)$$\begin{array}{c} \left\{\;\frac{\xi_{2}}{1\;+\;\xi_{2}}<a_{2}<\frac{\xi_{2}\left(1\;+\;\xi_{1}\right)}{\xi_{1}\;+\;\xi_{1}\xi_{2}\;+\;\xi_{2}}\;\right\}\;\cap \;\left\{\;\frac{\xi_{1}}{\xi_{1}\;+\;\xi_{1}\xi_{2}\;+\;\xi_{2}}<a_{1}<\frac{1}{1\;+\;\xi_{2}}\;\right\} \end{array}$$

**Remark 1.** 
*An arbitrary $\eta_{\pi (2)}=\frac{\xi_{1}}{a_{1}-\xi_{1}a_{2}}$ in power allocation range I-A has an equal $\eta_{\pi (1)}=\frac{\xi_{2}}{a_{2}-\xi_{2}a_{1}}$ in power allocation range II-A.*


#### 4.1.3. Infeasible
Power Allocation Range III

According to (Equation 8) and (Equation 11), there exists an infeasible power allocation range where $\xi_{1}\xi_{2}\geq 1$ ($\frac{\xi_{1}}{1+\xi_{1}}\geq \frac{1}{1+\xi_{2}}$ or $\frac{1}{1+\xi_{1}}\leq \frac{\xi_{2}}{1+\xi_{2}}$), making the denominator of $\eta_{\pi (i)}$ in Theorem 1 meaningless. We denote the infeasible power allocation range as the power allocation satisfying the inequality as follows:(14)$$\begin{array}{c} \left\{\frac{1}{1\;+\;\xi_{2}}<a_{1}<\frac{\xi_{1}}{1+\xi_{1}}\right\}\cap \left\{\frac{1}{1\;+\;\xi_{1}}<a_{2}<\frac{\xi_{2}}{1\;+\;\xi_{2}}\right\} \end{array}$$

Conversely, if $\xi_{1}\xi_{2}<1$ ($\frac{\xi_{1}}{1+\xi_{1}}<\frac{1}{1+\xi_{2}}$ or $\frac{1}{1+\xi_{1}}>\frac{\xi_{2}}{1+\xi_{2}}$), there is no infeasible power allocation range. Thus, feasible range I and feasible range II are stitched together seamlessly or even overlap partly.

**Remark 2.** 
*When $\xi_{1}\xi_{2}\geq 1$, there exists an unfeasible power allocation range, which is just a part of all the possible power allocation ranges. It can be completely skipped when finding the optimal power allocation. The optimal power allocation exists in the remaining possible power allocation ranges.*


### 4.2. Multi-User NOMA Case

According to permutation theory, *K* users have K! possible decoding orders, where (·)! is the factorial operator. We rewrite $\eta_{\pi (i)}=\max\left\{\frac{\xi_{\pi (l)}}{a_{\pi (l)}-\sum_{j=l+1}^{K}\xi_{\pi (l)}a_{\pi (j)}}\right\}$ as $\eta_{\pi (i)}=\max\left\{\eta_{\pi (i-1)},\frac{\xi_{\pi (i)}}{a_{\pi (i)}-\sum_{j=i+1}^{K}\xi_{\pi (i)}a_{\pi (j)}}\right\}$, where i=1,2,…,K,andl=1,2,…,i. Then, the feasible power allocation range under each decoding order can be divided into $2^{K-1}$ sub-ranges.

**Definition 1.** 
*The sub-range r-A in feasible power allocation range r is defined as the sub-range satisfying*

(15)
$$\begin{array}{c} \eta_{\pi (i)}=\frac{\xi_{\pi (i)}}{a_{\pi (i)}-\sum_{j=i+1}^{K}\xi_{\pi (i)}a_{\pi (j)}}. \end{array}$$



It can be deduced from (Equation 15) that $\eta_{\pi (i-1)}\leq \eta_{\pi (i)}$. Similar with the analysis of the two-user NOMA case, we will show the power allocation range *r*-A first for the multi-user NOMA case, r=I,II,…,K!. Denote $v_{\pi (i)}$ as the demarcation value of power allocation $a_{\pi (i)}$ in sub-range *r*-A. The demarcation values should satisfy $\eta_{\pi (1)}\leq \eta_{\pi (2)}\leq \ldots \leq \eta_{\pi (K)}$, which can be processed as
(16)$$\begin{array}{c} -\xi_{\pi (i+1)}v_{\pi (i)}+\xi_{\pi (i)}\left(1+\xi_{\pi (i+1)}\right)v_{\pi (i+1)}\leq 0,\;1\leq i,i+1\leq K. \end{array}$$

Recall that $\sum_{1}^{K}a_{\pi (i)}=1$. Then, we write an equation set as follows:(17)$$\begin{array}{c} \left\{\begin{array}{c} -\xi_{\pi (i+1)}v_{\pi (i)}+\xi_{\pi (i)}\left(1+\xi_{\pi (i+1)}\right)v_{\pi (i+1)}=0,\;1\leq i,i+1\leq K,\\ \sum_{1}^{K}v_{\pi (i)}=1. \end{array}\right. \end{array}$$

The demarcation values can be calculated using the equation set. In fact, the equation set can be written into a *K*-dimension matrix with *K* variables, which can be solved in Matlab 7.0 and more recent versions. Finally, we obtain sub-range *r*-A as follows:(18)$$\begin{array}{c} \left\{\begin{array}{c} v_{\pi (1)}\leq a_{\pi (1)}<1,\\ \ldots \\ \xi_{\pi (i)}\left(1+\xi_{\pi (i)}\right)v_{\pi (i+1)}/\xi_{\pi (i+1)}\leq a_{\pi (i)}\leq v_{\pi (i)},\\ \ldots \\ 0<a_{\pi (K)}\leq v_{\pi (K)}. \end{array}\right. \end{array}$$

It is clear that there are no CSI parameters in (Equation 17) or (Equation 18), i.e., the optimal power allocation ranges are only affected by the target data rates and total power. In a later section, we prove that sub-range *r*-A is the optimal power allocation range. For space reasons, we cannot display all of the analysis details of the useless $2^{K-1}-1$ sub-ranges one by one in each power allocation range *r*.

## 5. Sum Throughput Maximization Problem

Sum throughput in NOMA systems is defined as the sum of each target data rate times its respective probability of a successful decoding [1]. Let $T=\sum_{k=1}^{K}R_{k}^{F}\left(1-P_{k}^{\mathrm{out}}\right)$ denote the maximum sum throughput. Now, we formulate a sum throughput maximization problem as follows
(19a)$$\max_{\pi ,a}\;\;T$$
(19b)$$\;\;\;\;\;\;\;\;\;\;\;\;\;s.t.\;\;a_{\pi (i)}\in \mathrm{Range}\;r-A,r=1,\ldots K!$$
(19c)$$\;\;\;\;\;\;\sum_{i=1}^{K}a_{\pi (i)}=1.$$

**Theorem 4.** 
*For an arbitrary decoding order in a K-user case, the maximum sum throughput is always achieved in the optimal power allocation range r-A, where r=I,II,…,K!,K≥2.*


**Proof.** Let $\eta_{\pi (i)}^{*}=\frac{\xi_{\pi (i)}}{a_{\pi (i)}^{*}-\sum_{j=i+1}^{K}\xi_{\pi (i)}a_{\pi (j)}^{*}}$ and $\eta_{\pi (i+1)}^{*}=\frac{\xi_{\pi (i+1)}}{a_{\pi (i+1)}^{*}-\sum_{j=i+1}^{K}\xi_{\pi (i+1)}a_{\pi (j)}^{*}}$. We set power allocation $a^{*}$ from power allocation sub-range *r*-A to guarantee $\eta_{\pi (i+1)}^{*}\geq \eta_{\pi (i)}^{*}$ on the condition as shown in (Equation 18). Similarly, let $\eta_{\pi (i)}^{\prime }=\frac{\xi_{\pi (i)}}{a_{\pi (i)}^{\prime }-\sum_{j=i+1}^{K}\xi_{\pi (i)}a_{\pi (j)}^{\prime }}$ and $\eta_{\pi (i+1)}^{\prime }=\frac{\xi_{\pi (i+1)}}{a_{\pi (i+1)}^{\prime }-\sum_{j=i+1}^{K}\xi_{\pi (i+1)}a_{\pi (j)}^{\prime }}$. We set another power allocation $a^{\prime }$ from another power allocation sub-range *r*-B to meet $\eta_{\pi (i+1)}^{\prime }\leq \eta_{\pi (i)}^{\prime }$. Thus, for arbitrary $\eta_{\pi (i+1)}^{\prime }$ from sub-range *i*-B, we can always find an equal $\eta_{\pi (i+1)}^{*}$ from sub-range *i*-A, i.e., $\eta_{\pi (i+1)}^{*}=\eta_{\pi (i+1)}^{\prime }$. Then, we have the corresponding relationship $\eta_{\pi (i)}^{*}\leq \eta_{\pi (i)}^{\prime }$. It is known that $P_{\pi (i)}^{\mathrm{out}}\left(x\right)$ is a monotonically increasing function about variable *x*. If $\eta_{\pi (i)}^{*}\leq \eta_{\pi (i+1)}^{*}=\eta_{\pi (i+1)}^{\prime }\leq \eta_{\pi (i)}^{\prime }$, then $P_{\pi (1)}^{\mathrm{out}}\left(\eta_{\pi (i)}^{*}\right)\leq P_{\pi (1)}^{\mathrm{out}}\left(\eta_{\pi (i)}^{\prime }\right)$ and $\;P_{\pi (2)}^{\mathrm{out}}\left(\eta_{\pi (2)}^{*}\right)=P_{\pi (2)}^{\mathrm{out}}\left(\eta_{\pi (2)}^{\prime }\right)$ Now, we obtain the inequality $R_{\pi (i)}^{F}P_{\pi (i)}^{\mathrm{out}}\left(\eta_{\pi (1)}^{*}\right)+R_{\pi (i+1)}^{F}P_{\pi (i+1)}^{\mathrm{out}}\left(\eta_{\pi (i+1)}^{*}\right)\leq R_{\pi (i)}^{F}P_{\pi (i)}^{\mathrm{out}}\left(\eta_{\pi (i)}^{\prime }\right)+R_{\pi (i+1)}^{F}P_{\pi (i+1)}^{\mathrm{out}}\left(\eta_{\pi (i+1)}^{\prime }\right)$. So, power allocation $a^{*}$ from sub-range *r*-A can always provide greater sum throughput than that from other sub-ranges under the same decoding order. □

According to Theorem 4, we only need to search in optimal power allocation sub-range *r*-A under each decoding order to solve the optimization problem, where r=I,II,…,K!. The flow chat in Figure 1 below helps to express our method for solving the optimization problem. It is worth noting that both the optimal decoding order and optimal power allocation remain unchanged as long as the statistical CSI remains unchanged.

## 6. Simulations

We conducted Monte Carlo simulations (using $10^{6}$ realizations) to verify the theoretical expression of our derived exact outage probability. Take a two-user case as an example. Set $\sigma_{1}^{2}=\sigma_{2}^{2}=1$. Other related simulation parameters are given in the blank area on each figure for convenience, e.g., $R^{F}=\left[R_{1}^{F}\;R_{2}^{F}\right]$ shows the target data rates of $U_{1}$ and $U_{2}$, respectively.

Figure 2 shows the outage probability versus the SNR at a fixed power allocation. Monte Carlo (MC) simulations well verified the accuracy of our theoretical exact outage probability. Our derived asymptotic (Asym) outage probability clearly expresses the characteristic of the exact outage probability in the high-SNR regime. The slope of the asymptotic outage probability represents the diversity order.

Figure 3 depicts all the power allocation ranges for two users according to the analysis in Section 4.1, where $p_{2}=a_{2}P$ and $p_{1}=a_{1}P$ are the labels of the X-axis and Y-axis, respectively. We mark $M_{1}\left(0,P\right)$ and $V_{1}\left(\frac{P}{1+\xi_{1}},\frac{\xi_{1}P}{1+\xi_{1}}\right)$ as the two demarcation points of power allocation range I, and $V_{2}\left(\frac{\xi_{2}P}{\xi_{1}+\xi_{1}\xi_{2}+\xi_{2}},\frac{\xi_{1}\left(1+\xi_{2}\right)P}{\xi_{1}+\xi_{1}\xi_{2}+\xi_{2}}\right)$ is the demarcation point between power allocation range I-A and power allocation range I-B. Similarly, we mark $M_{2}\left(P,0\right)$ and $V_{3}\left(\frac{\xi_{2}P}{1+\xi_{2}},\frac{P}{1+\xi_{2}}\right)$ as the two demarcation points of power allocation range II, and $V_{4}\left(\frac{\xi_{2}\left(1+\xi_{1}\right)P}{\xi_{1}+\xi_{1}\xi_{2}+\xi_{2}},\frac{\xi_{1}P}{\xi_{1}+\xi_{1}\xi_{2}+\xi_{2}}\right)$ is the demarcation point between power allocation range II-A and power allocation range II-B. It can be found that the demarcation points are affected by the target data rates and total power, with no effect from the CSI. And the values of the demarcation points are proportional to the total power. In considering $\mathrm{SNR}=P/\sigma^{2}$, this figure also shows the effect of the SNR on the power allocations when we fix $\sigma^{2}$.

Figure 4 sketches the exact outage probability of two users under different power allocation ranges. It shows that power allocation sub-range I-A outperforms sub-range I-B under the same decoding order π=(1,2). Similarly, sub-range II-A outperforms sub-range II-B under the same decoding order π=(2,1). Therefore, we just need to search in optimal power allocation sub-ranges I-A and II-A for the sum throughput maximization problem. Interestingly, the outage probability curve of range I-B perfectly overlaps with that of range II-B, which agrees well with our *Remark 1*. It is worth noting that the outage probabilities of the demarcation points $V_{2}$ and $V_{4}$ in Figure 3 are located identically at the same turning point marked by “o” in this figure. This indicates the accuracy of our theoretical analysis of demarcation points $V_{2}$ and $V_{4}$.

Figure 5 shows the maximum sum throughput values achieved by the optimal decoding order $\pi^{*}$ versus the total transmit power. The maximum sum throughput curves under decoding order π=(1,2) and π=(2,1) intertwine together in different power regimes. Therefore, there does not exist a changeless optimal decoding order such that its corresponding maximum sum throughput is always greater than another one in all the power regimes. The method used to decide the optimal decoding order was to compare the maximum sum throughput values under different decoding orders. We selected the decoding order that produced the greater maximum sum throughput as the optimal decoding order.

## 7. Discussion

As we stated in the motivation part, well-designed searching algorithms, including machine learning algorithms, can be combined with our reduced power allocation range to improve the searching efficiency. This is the main advantage of our work. Without our theoretically optimal power allocation range, the searching algorithms previously proposed by researchers, such as those mentioned in the introduction part, need to search in the whole problem definition domain.

However, we just considered a single antenna at each node in this study. This is the obvious limitation of our work. We will try to conduct more meaningful work in future research by extending the proposed model to multiple antennas. For example, there exist some open problems about the optimal power allocation ranges considering beamforming or antenna selection. Our proposed theoretical analysis method used to achieve optimal power allocation and the joint optimization method in this paper were based on statistical CSI, whereas a majority of studies using beamforming usually consider perfect CSI. Thus, the extension to beamforming is quite challenging. In addition, it is also worth pondering how the increase in channel complexity affects the use of our proposed optimization method. When multiple antennas are used as nodes, the channel complexity will increase. Consider a downlink MISO-NOMA system as a simple example. If the antenna selection technique is adopted at the BS, the outage probability used in (Equation 5) will change, but the optimal power allocation range in Theorem 4 is still suitable.

## 8. Conclusions

We studied a single-cell downlink NOMA system considering statistical CSI. The closed-form expressions of the exact outage probability were derived. The asymptotic outage probability and diversity order were also deduced for insight on the characteristic of the outage probability in a high-SNR regime. In thoroughly analyzing the possible power allocation ranges, the optimal power allocation range under a certain decoding order was obtained in a theoretical form. The demarcation points of the optimal power allocation ranges are affected by the target data rates and total power, with no effect from the CSI. In particular, the values of the demarcation points are proportional to the total power. A sum throughput maximization problem was formulated, and it was solved by efficiently searching in our achieved optimal power allocation ranges. The numerical results show that our deduced demarcation points of the optimal power allocation ranges are accurate, and the optimal decoding order is not constant at different total transmit power levels.

## Figures and Tables

**Figure 1 entropy-26-00421-f001:**
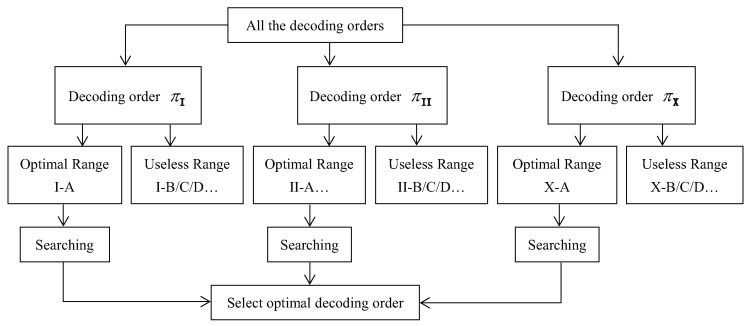
Method to solve the optimization problem. Each decoding order has an optimal power allocation sub-range, and the optimal decoding order is the one that produces the maximum sum throughput.

**Figure 2 entropy-26-00421-f002:**
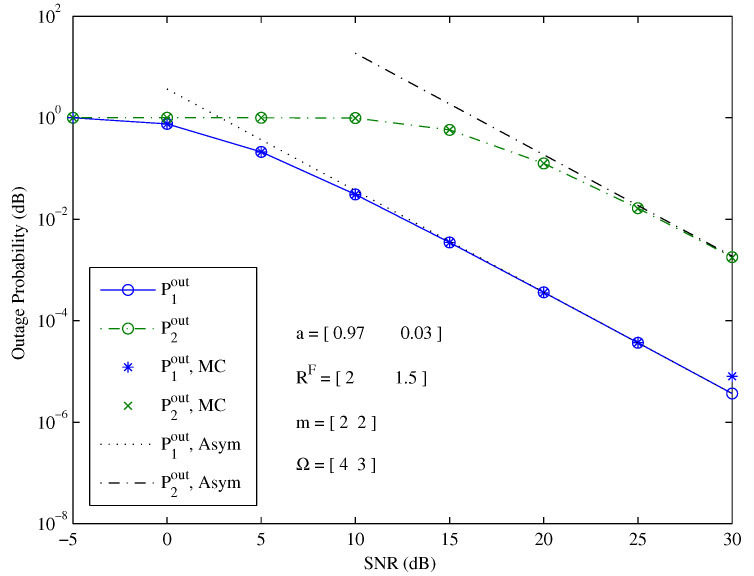
The outage probability of two users under a given decoding order and fixed power allocation.

**Figure 3 entropy-26-00421-f003:**
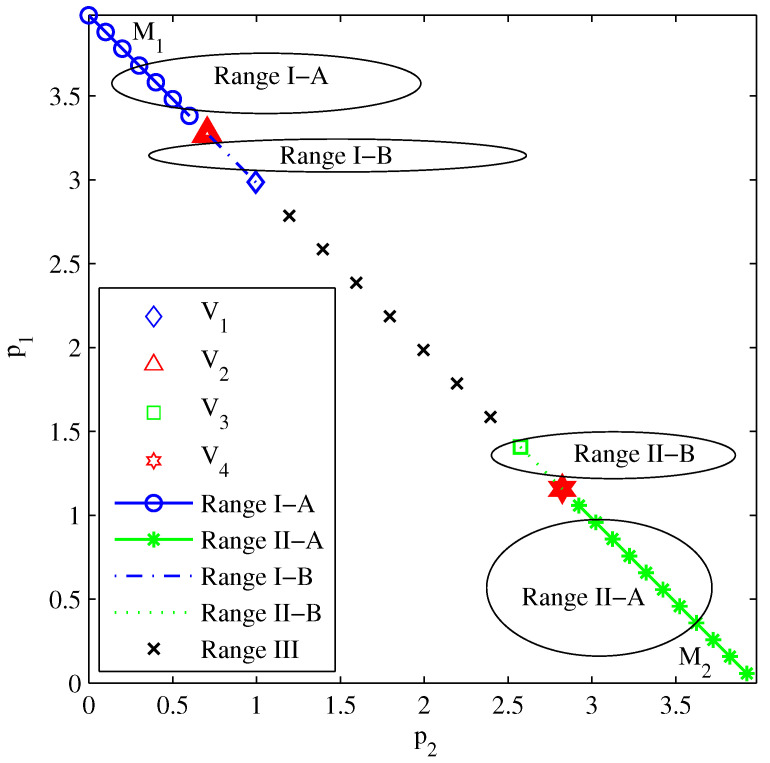
All the power allocation ranges for two-user NOMA.

**Figure 4 entropy-26-00421-f004:**
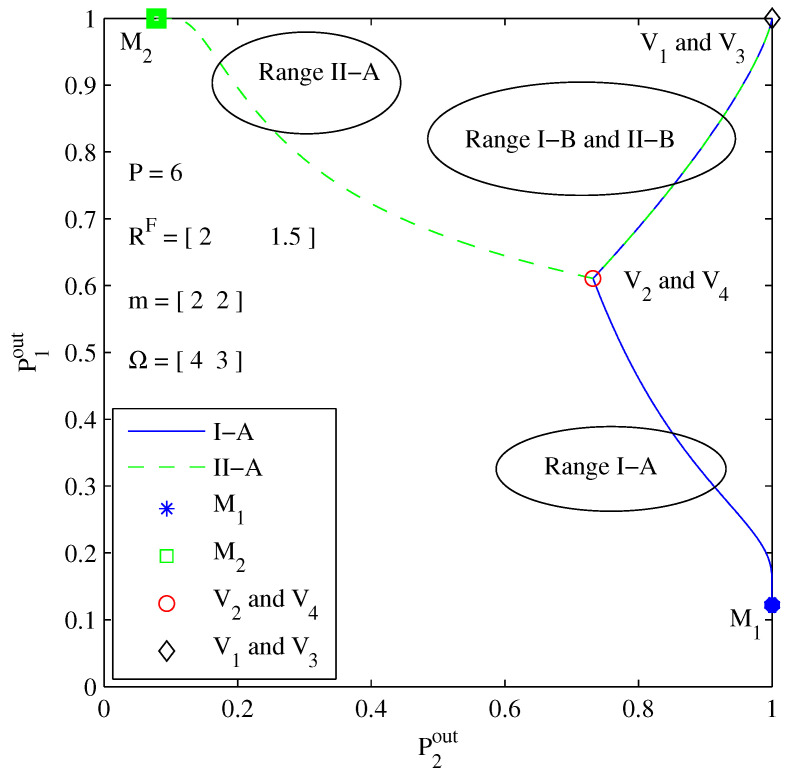
Outage probability scatter diagram of two users under different power allocation ranges.

**Figure 5 entropy-26-00421-f005:**
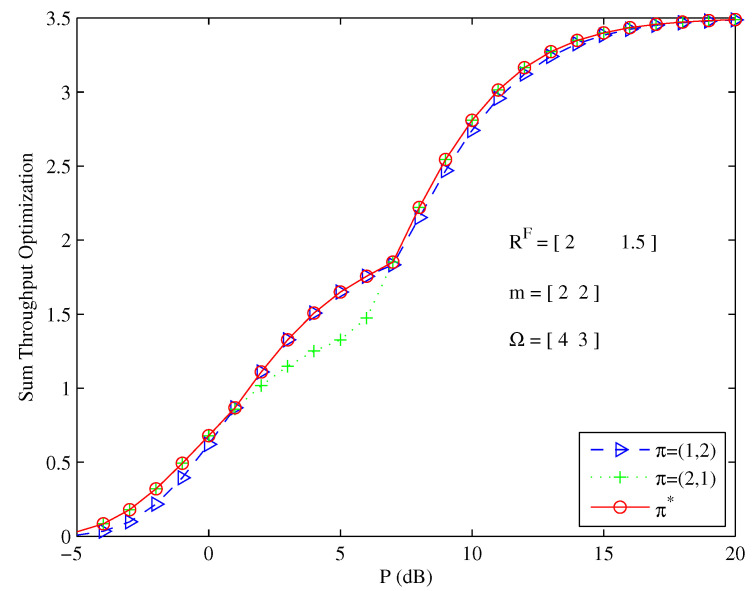
Optimal decoding order for sum throughput maximization problem at different total power levels.

## Data Availability

Data are contained within the article.

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
