# Peer review of "Optimal Decoding Order and Power Allocation for Sum Throughput Maximization in Downlink NOMA Systems"

_entropy, 2024, doi:10.3390/e26050421_

Round 1

Reviewer 1 Report

Comments and Suggestions for Authors

 A derivation of the optimal decoding order for a two-user NOMA system is presented in this manuscript. There are a number of issues with the approach followed in the paper:

1) While all the derivations and calculations are correct, the proposed model is rather simplistic, since it only considers two users (so only two decoding orderings are possible) and the performance formulas are relatively easy to analyze. Compared to the vast literature on NOMA and its possible configurations, it is not clear what this model contributes on for a rather limited setting.

2) Starting at equation (6) and later on, the authors introduce implicitly the assumption that the expectation of the Nakagami statistics is an integer. While this is not a significant loss of generality, since the asymptotic regime is not changed, it should be better motivated and explained.

3) The main weakness is, however, the specificity of the presented model, with two single-antenna users only. Even if this is analyzed to get some insights, it is hard to see how to generalize the obtained results to an arbitrary number of users, unless some ordering pin the statistics of the channel quality is imposed.

Overall, the formulation is correct and so are the calculations. But the results presented here are somewhat limited in spirit, in my opinion, and shed little insight into more complex systems. Also, the model is particularly built for Nakagami channels under some tractable conditions and assumption. So, probably, the main problem is that the extension to more users and/or multiple antennas is not addressed, which falls a little too short for my view of the problem.

Comments on the Quality of English Language

The English language used in the paper is correct.

Reviewer 2 Report

Comments and Suggestions for Authors

Reviewer's report

Manuscript ID: entropy-2874136

Title: Optimal Decoding Order and Power Allocation for Sum Throughput Maximization in Downlink NOMA Systems networks

Zhuo Han 1, Wanming Hao 1, Zhiqing Tang 2, Shouyi Yang 1

Paper idea:

The authors propose a solution approach for exact outage probability according to given decoding orders in downlink NOMA-based networks. They have also provided an optimization method for throughput maximization.

The outcomes of the proposed cross layer is simulation-based results and releveled that the system throughput is improved and the edge users are satisfied.

The authors propose a scheme for vehicular communications based on Q-L-machine

Strength:

-The paper is well written as well as well mathematically analyzed

Weakness:

- The authors should explain what is the novelty of this work compared to the related works wherein there are plenty of works have discussed that topic

-The authors may provide a flow chart to show the decoding orders and its relation with power allocations and of the proposed scheme is not clear in the presented table.

- The effect of channel state information (CSI), is not clear

-The outcomes of the proposed work is good but not enough, the authors may provide some results to study the effect of SNR in power allocations, the effect on the outage probability too. Also, other studying may be provided to see the accuracy of the provided expression.

Reviewer 3 Report

Comments and Suggestions for Authors

This paper analyzes a single-cell downlink NOMA system considering statistical CSI. The outage probability and the power allocation scheme are studied. A major revision is recommended.

Q1. Only the situation of two users are studied. The author should explain which is the barrier of more users?

Q2. The Nakagami-m channels are adopted. What is the difference between the Rayleigh channels? The motivation of Nakagami-m channels is not clear.

Q3. There are two ways of calculating the diversity order, which are BER and outage probability. Why the authors did not calculate the diversity order by the BER?

Q4. This paper only analyzes the PD-NOMA. Why CD-NOMA are not given as a compassion?

Q5. While the references are current and pertinent, they could be further improved to encompass a broader aspect of the subject matter, thereby enhancing the rigor and quality of the research. This expansion would also make them more applicable and suitable for this work. Readers would greatly benefit from the inclusion of referenced research articles in the introduction.

[1]Li, Y.; Chen, Y.; Ji, X. Secure User Pairing and Power Allocation for Downlink Non-Orthogonal Multiple Access against External Eavesdropping. Entropy 2024, 26, 64. https://doi.org/10.3390/e26010064

[2] Zhou, X. An Efficient Block Successive Upper-Bound Minimization Algorithm for Caching a Reconfigurable Intelligent Surface-Assisted Downlink Non-Orthogonal Multiple Access System. Electronics 2024, 13, 791. https://doi.org/10.3390/electronics13040791

[3] S. Guo, X. Zhao and W. Zhang, "Throughput Maximization for RF Powered Cognitive NOMA Networks with Backscatter Communication by Deep Reinforcement Learning," in IEEE Transactions on Wireless Communications, doi: 10.1109/TWC.2023.3337409.

Reviewer 4 Report

Comments and Suggestions for Authors

General comments:

The presented analysis and simulation results show a strict dependence of the total throughput on the total power of the two receiving stations. The authors presented a simple approach to modeling the NOMA technique, as low channel variability and simple antenna systems were assumed. A short references study was also performed to show that authors’ considerations constitute a significant extension of research conducted by other scientists. The reviewer has no major objections to the content and language of the article (version 3). After a small expansion of selected issues, the work can be successfully published.

Detailed comments:

The paper was written in a concise and readable way. I propose introducing several extensions to increase the attractiveness of the article (especially since the descriptions of the results obtained are too synthetic):

1. Please discuss changes to the model when sector or smart beamforming antennas are used.

2. I suggest that after the chapter on simulations, a chapter on the discussion of the results with reference to those obtained by other scientists be added to indicate how the proposed analytical improvements can be combined with existing techniques (proposed by other scientists).

3. Can machine learning be used in searching for optimal solutions combining the elements listed in subsection 1.1 with those proposed in the article?

4. How will the increase in channel complexity affect the use of the proposed optimization method?  An extended case study is needed, which should also be included in the discussion chapter.

Round 2

Reviewer 1 Report

Comments and Suggestions for Authors

The authors have extended their model to cover the case with K > 2 receivers, but there still persists the assumption that the decoding order is unique for all the receivers. Though downlink NOMA is certainly more complex than uplink NOMA in this respect, the finding that there is un unfeasible power allocation (under statistical CSI information) comes directly from this assumption.

I think the paper formulates an interesting problem, but this problem is possibly too far from a realistic setting, since (i) statistical CSI impacts only on the outage probability, as expected; (ii) a fixed decoding order does not fully exploit all the design space for the problem.

Comments on the Quality of English Language

The quality of the English language used in the manuscript needs to be improved, e.g., 'technic' -> technique / technical (as a noun / adjective). Other parts include unfinished sentences or sentences without a definite meaning.

Reviewer 2 Report

Comments and Suggestions for Authors

the authors have answered all comments

Comments on the Quality of English Language

The English of the paper is good

Author Response

Thank you so much.

Reviewer 3 Report

Comments and Suggestions for Authors

The authors have answered all my comments. The revised paper may be accepted in the present form.

Author Response

Thank you so much.

Round 3

Reviewer 1 Report

Comments and Suggestions for Authors

I appreciate the effort made by the authors in improving and clarifying some points in the revised version, but the paper still suffers from a limited contribution, in my opinion. This version includes some minor changes into the Introduction, basically, but the technical core of the paper remains the same and with the same focus. While the paper is technically correct in the model and derivations, the calculations and conclusions are relatively straightforward.
